# Quantification of droplet and contact transmission risks among elementary school students based on network analyses using video-recorded data

Shuta Kikuchi[1], Keisuke Nakajima[1], Yasuki Kato[1], Takeshi Takizawa[1], Junichi Sugiyama[1], Taisei Mukai[2], Yasushi Kakizawa[1], Setsuya Kurahashi[3]*

1 Advanced Analytical Science Research Laboratories, Research & Development Headquarters, Lion Corporation, Edogawa, Tokyo, Japan, 2 Institute of Social Simulation, Waseda University, Shinjuku, Tokyo, Japan, 3 Graduate School of Business Science, University of Tsukuba, Bunkyo, Tokyo, Japan

☯ These authors contributed equally to this work.
* kurahashi.setsuya.gf@u.tsukuba.ac.jp

## Abstract

In elementary schools, immunologically immature students come into close contact with each other and are susceptible to the spread of infectious diseases. To analyze pathogen transmission among students, it is essential to obtain behavioral data. Questionnaires and wearable sensor devices were used for communication behavior and swab sampling was employed for contact behavior. However, these methods have been insufficient in capturing information about the processes and actions of each student that contribute to pathogen transmission. Therefore, in this study, actual behavioral data were collected using video recordings to evaluate droplet and contact transmission in elementary schools. The analysis of communication behavior revealed the diverse nature of interactions among students. By calculating the droplet transmission probabilities based on conversation duration, the risk of droplet transmission was quantified. In the contact behavior, we introduced a novel approach for constructing contact networks based on contact history. According to this method, well-known items, such as students' desks, doors, and faucets, were predicted to be potential fomite. In addition, students' shirts and shared items with high contact frequency and high centrality metrics in the network, which were not evaluated in swab sampling surveys, were identified as potential fomites. The reliability of the predictions was demonstrated through micro-simulations. The micro-simulations replicated virus transmission scenarios in which virus-carrying students were present in the actual contact history. The results showed that a significant amount of virus adhered to the items predicted to be fomites. Interestingly, the micro-simulations indicated that most viral copies were transmitted through single items. The analysis of contact history, contact networks, and micro-simulations relies on video-recorded behavioral data, highlighting the importance of this method. This study contributes significantly to the prevention of infectious diseases in elementary schools by providing evidence-based information about transmission pathways and behavior-related risks.

**Data Availability Statement:** All relevant data are within the manuscript and its Supporting information files.

**Funding:** This study was financially supported by JSPS KAKENHI (https://www.jsps.go.jp/) in the form of grants (21H01561 and 23H00503A) received by S. Kurahashi. This research was also supported by a joint research fund from Lion Corporation (https://www.lion.co.jp/) in the form of an award (CII03136) received by S. Kurahashi. The funders had no role in study design, data collection and analysis, decision to publish, or preparation of the manuscript.

**Competing interests:** The authors have read the journal's policy and have the following competing interests: S. Kikuchi is an employee of Lion Corporation. KN is an employee of Lion Corporation. Y. Kato is an employee of Lion Corporation. TT is an employee of Lion Corporation. JS is an employee of Lion Corporation. Y. Kakizawa is an employee of Lion Corporation. There are no patents, products in development or marketed products associated with this research to declare. This does not alter our adherence to PLOS ONE policies on sharing data and materials.

## Introduction

Human respiratory viruses cause significant morbidity, mortality, and economic losses globally each year [1–4]. Occasional pandemics, such as the COVID-19 pandemic and the 2009 H1N1 influenza pandemic, have severely disrupted society and economics.

Viruses can be transmitted primarily via two pathways [5, 6]. The first is "droplet transmission," which occurs through exposure to droplets produced by coughing, sneezing, and conversations with virus carriers. Based on their size, these droplets can be further categorized into large droplets and fine aerosols. The second pathway is "contact transmission," which occurs when a virus carrier comes into contact with surfaces contaminated with droplet-borne viruses. Contact transmission can be further categorized into direct contact, which involves direct physical contact, and indirect contact, which involves contact with fomites.

To estimate virus transmission risk in elementary schools, it is essential to obtain behavioral data from students that detail who was involved, when, where, and what activities were performed, particularly concerning droplet and contact transmissions. However, existing methods have been insufficient in capturing information about the processes and actions of each student that contribute to pathogen transmission.

In this study, we aim to accurately assess the comprehensive pathogen transmission risk within classroom environments in elementary schools by using video-recorded data to observe students' behaviors in real-time. By analyzing communication and contact networks among students and using a micro-simulation model, we assessed the risk of both droplet and contact transmission in elementary schools. Using video recording, we were able to obtain the processes and actions of each individual student. This approach enables the identification of potential factors contributing to both droplet and contact transmission risks. Regarding communication behavior, networks among students and the duration of conversations were investigated. Droplet transmission probability was calculated based on the cumulative conversation duration for each pair of students. Regarding contact behavior, the contact items and their frequencies were investigated. In addition, we developed a novel analysis method that constructed networks based on contact history. Fomites that mediated virus transmission were predicted through a network analysis. These predictions were confirmed through a micro-simulation that simulated virus transmission based on the actual contact history. The micro-simulations indicated that the majority of virus copies were transmitted through single items. The network analysis and micro-simulations based on contact history were made possible through detailed contact behavioral data obtained from video recordings. Hence, this study contributes to the understanding of droplet and contact transmission in elementary schools and provides insights into infection prevention.

## Related work

Droplet and contact transmissions typically spread through social networks [7–10]. Schools are an important environment for virus transmission because of the close contact between students, teachers, and school staff [11–16]. Furthermore, infected elementary school students can become an infection source within their households, spreading the infection throughout the community [12, 17–19]. Therefore, investigating virus transmission in elementary schools is important for preventing outbreaks.

To estimate virus transmission in elementary schools, various methods have been employed to collect infection-related behavioral data.

The evaluation of droplet transmission used the analysis of communication between students. Questionnaires and wearable sensor devices were used for communication behavior [11, 16, 20–24]. Mikolajczyk et al. [11] collected communication data from elementary school

students in Germany using questionnaires. The distribution of daily communication numbers was analyzed based on age and differences between weekdays and weekends. In addition, a survey on illness within the last six months was conducted with the participants. Statistical analysis suggested that the risk of infection differed by an odds ratio of 2.45 depending on the presence or absence of siblings. However, a questionnaire cannot capture details such as the type and duration of communication in actual behaviors. Guclu et al. [23] analyzed communication in schools of the United States, including elementary, elementary-middle, middle, and high schools, using wireless sensor devices that regularly recorded other devices within a distance of 3 meters. In addition to the communication frequencies, they analyzed communication duration and demonstrated that these followed a power-law distribution. Furthermore, they analyzed communication networks across different classes and grades. This result identified that communication was more frequent within the same grade or class, except for high schools. Guo et al. [24] collected behavioral data from students in China using wearable devices that recorded real-time close contact behaviors, including interpersonal distance, facial orientation, and relative positions of students. Their findings revealed that close contact rates were higher during breaks than during classes. During breaks, short-range airborne transmission routes became more significant. However, wearable sensor devices can only detect proximal communication and cannot determine whether a conversation took place. Grantz et al. [16] collected communication interactions among students aged 5 to 18 years in the United States using questionnaires and wearable sensor devices. The questionnaire survey reported fewer communication interactions, but they tended to involve longer durations. In contrast, the wearable sensors recorded shorter communication durations, highlighting differences in results based on the measurement method.

In contact behavior, it is difficult to obtain information about contacted items and the order and frequency thereof through questionnaires and wearable sensor devices. Hence, research on contact transmission in elementary schools was limited to methods that detect pathogens present on the surfaces of school items through swab sampling [25–29]. Bright et al. [25] investigated the presence of bacteria, norovirus, and influenza A virus on the surfaces of items in elementary school classrooms. Items with high contact frequency, such as faucets, desks, pencil sharpeners, keyboards, and paper towel dispensers, were found to be the most contaminated with bacteria and viruses. Fong et al. [26] performed swab sampling in elementary schools and kindergartens attended by students aged 4 to 9 years. Influenza virus was detected on bookshelves and doorknobs/door surfaces inside classrooms. Zulli et al. [29] performed swab sampling on desks in elementary schools. Rhinovirus, adenovirus, and norovirus were detected on the desks. While swab sampling provides insights into the outcomes of behaviors, it is incapable of capturing information about the processes leading to those outcomes. In addition, it is not possible to analyze the impact of individual students.

Thus, the methods used in previous studies have been insufficient in capturing detailed information about the processes and actions of each student that contribute to pathogen transmission.

## Methods

### Data collection/ethics statement and privacy

The methodology used in this study was approved by the Institutional Ethics Review Board of the University of Tsukuba and Lion Corporation. In preparation for this survey, approval was obtained from the principal of cooperating school on 22-10-2022. The recruitment period for this study began on 30-11-2022 and ended on 22-12-2022. An information letter, explaining the study was sent to the guardians of the participating students by the school principal at 30-

11-2022. Written consent was subsequently obtained from the guardians of the participants. From 30-11-2022 to 08-12-2022, the students and their guardians were allowed to opt out of the study. Furthermore, during the study period, until its conclusion on 22-12-2022, students and their guardians could withdraw their consent at any time. If consent was not given or was withdrawn, the respective guardians and students were instructed to submit a designated form, included in the information letter, to the teacher and the video recording team. The authors received only a report on the number of participants who opted out or withdrew consent.

The survey was conducted in a class at an elementary school in Tokyo, Japan, for four days in 12-2022. The students were 9–10 years of age and the class comprised 30 students. Communication and contact behaviors were video-recorded using four and three cameras set in the classroom and hallway, respectively. The videos recorded the time from the arrival to the departure from the elementary school. The break time, that is, before morning and afternoon homeroom, three 5-minute breaks, one 10-minute break, and a lunch break, excluding class hours, was used for the analyses. For privacy protection purposes, communication and contact behaviors were annotated from the recorded video data by a third-party institution. In addition, the video was pixelated after annotation by the third-party institution to prevent the identification of individuals. Each student was assigned an ID and analyzed anonymously once the data reached the researchers. Images of the contacted items (including faucets and hand wash) and the recorded classrooms, hallways, handwashing areas, and toilets, were not retained.

Communication behavior was annotated with the timestamp of the communication start time, the ID of the student who initiated communications (the "initiator"), the student who was engaged in communication with the initiator ("target"), and the communication duration. Communication was categorized into three types: conversation, contact (physical contact), and conversation and contact (both conversation and physical contact).

The annotation of the contact behavior included the student who contacted something, the contacted items including body parts (referred to as the "item"), and the owner of the item (either a personal belonging of a student or a shared common item). When a student contacted an item, it was recorded whether the individual was alone or in the presence of other people. These states were labeled "solo behavior" and "group behavior," respectively. The classroom and hallway layouts are presented in Fig 1. Each faucet and hand wash was annotated individually and the items located at the front, back and sides of the classroom were also indicated individually.

It should be noted that not all breaks, during which the videos were recorded, have been annotated.

## Network analyses

To analyze the communication and contact patterns among students, a network for each was generated.

The communication network was generated from the adjacency matrix of the initiators and targets. The communication network is represented as a directed graph, in which communication is conducted from the initiator to the target with a specific directionality. The degree of the communication network is the number of students who communicate during an arbitrary period. This value was counted if at least one communication was conducted between students ID:$i$ and ID:$j$. The in-degree and out-degree represent the number of initiators and targets, respectively. The number of communications between a pair of students, ID:$i$ and ID:$j$, was counted during an arbitrary period.

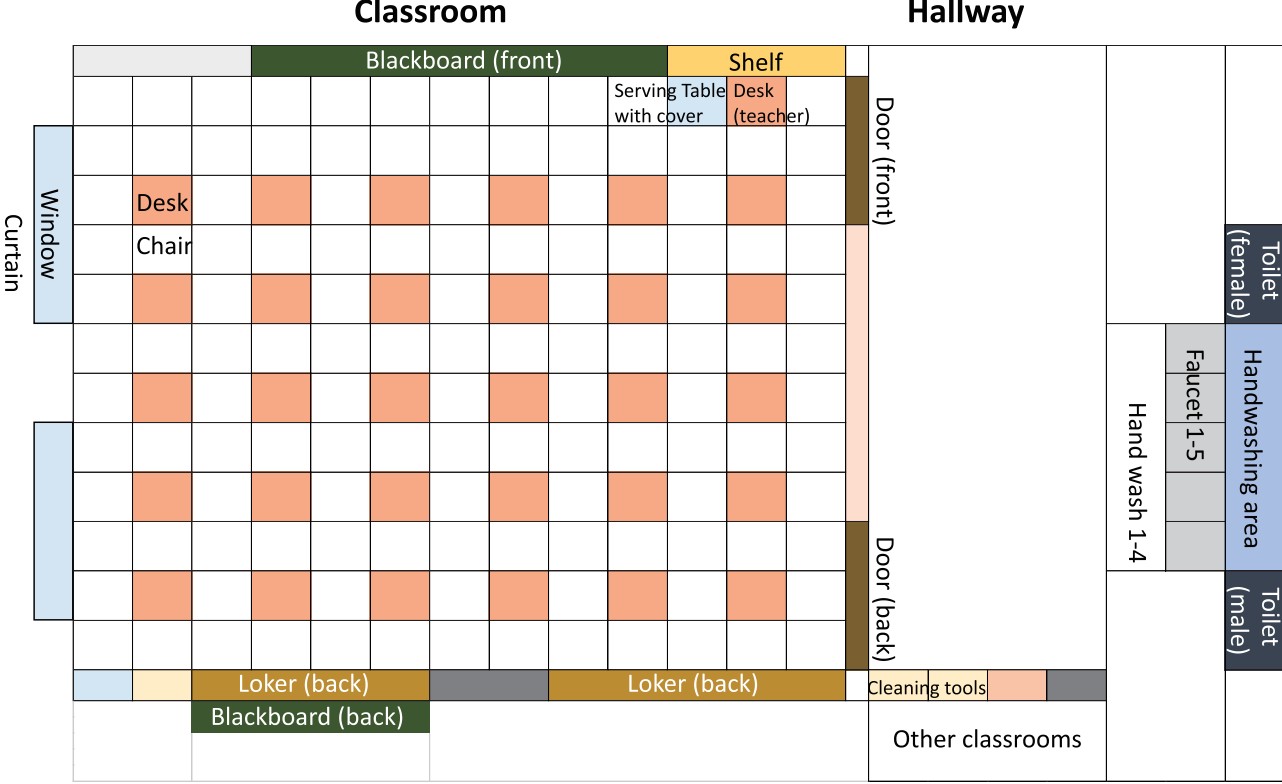

**Fig 1. Layout of measurement area.**

The contact network was represented as an undirected graph. When a student ID:$i$ contacted an item ID:$x$, an edge was generated between the student's hand ID:$i$ and the item ID:$x$. An example of a network is presented in Fig 2. The students' hands and items are represented by black and orange circles, respectively. The degree is the number of students who contacted an item during an arbitrary period. The shortest paths between the nodes of the student's hand were computed using Dijkstra's algorithm [30]. The betweenness centrality of node $c_b(v_i)$ was calculated as follows [31]:

$$c_b(v_i) = \sum_{v_s, v_t \in V_h} \frac{\sigma(v_s, v_t | v_i)}{\sigma(v_s, v_t)}, \tag{1}$$

where $V_h$ is the set of nodes of the student's hand, and $v_i$, $v_s$, and $v_t$ are the $i$-th, start, and target nodes of the student's hand, respectively. Moreover, $\sigma(v_s, v_t)$ is the number of shortest paths between $v_s$ and $v_t$, and $\sigma(v_s, v_t | v_i)$ is the number of those paths that pass through $v_i$ other than $v_s$ and $v_t$.

The degree distribution indicates the characteristics of the network. The probability $P(k)$ at degree $k$ is given by the following equation:

$$P(k) = \frac{1}{N} \sum_{i=1}^{N} \delta(k_i, k), \tag{2}$$

where $N$ is the number of nodes, and $k_i$ is the degree of the $i$-th node. Furthermore, $\delta(k_i, k)$ represents the Kronecker delta function, which equals one when $k = k_i$ and zero otherwise.

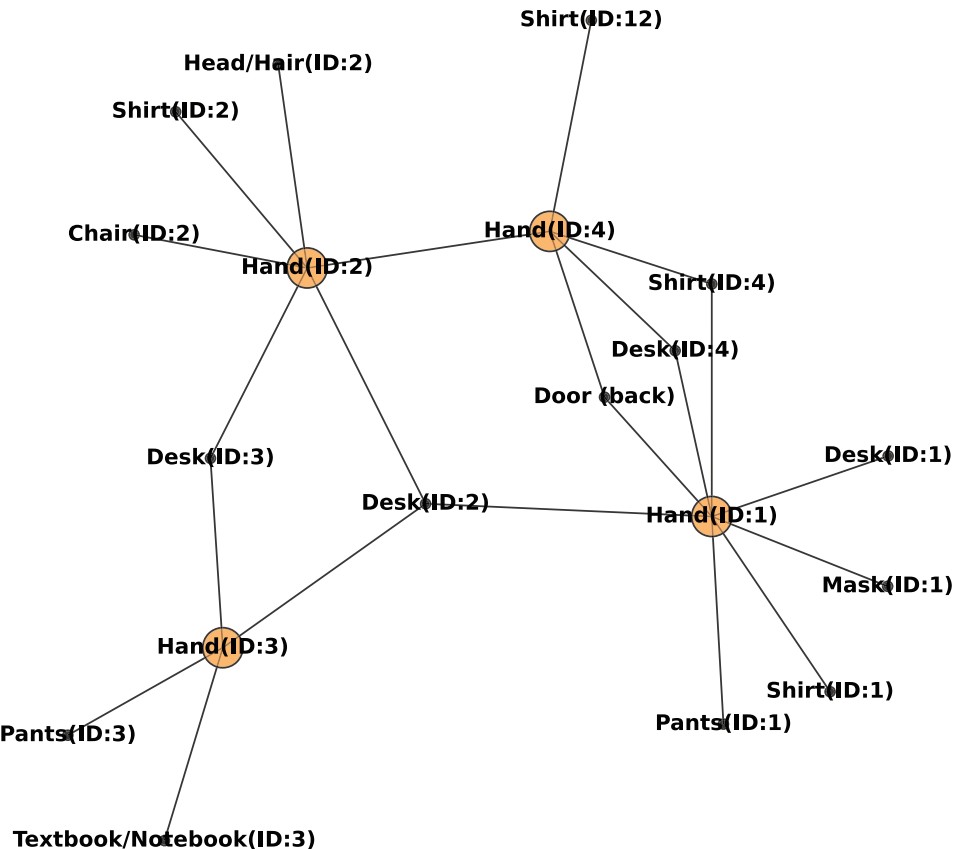

**Fig 2. Contact network example.** Orange and black circles represent students' hands and items, respectively.

To visualize these networks, the Fruchterman–Reingold force-directed algorithm [32] was employed to determine the positions of the nodes. These network analyses and visualizations were conducted using the `NetworkX` library [33].

## Probability of infection through conversation

The probability of droplet infection in susceptible students through conversation with virus-carrying students was calculated using an equation derived from numerical simulations [34–36].

The probability of droplet infection $P$ is given by [35, 37, 38]:

$$P = 1 - \exp\left(-\alpha \frac{N(x, T)}{N_0}\right), \tag{3}$$

where $\alpha$ is a factor that regulates the infectivity caused by viral strains. $N_0$ is assumed to be the average number of virus particles required to infect an individual. In this study, $N_0$ was set to values ranging from 300 to 2000, with a focus on 900. This value has been used as $N_0$ of SARS-CoV-2 in previous studies and falls within a range similar to that of influenza A [35, 36, 39]. The number of inhaled virons, denoted as $N(x, T)$, depends on the conversation duration $T$ and the distance $x$ between the students. The number of inhaled virons, $N(x, T)$, is represented

by

$$N(x, T) = \frac{B\lambda\bar{v}_0(x)T}{V_B}, \tag{4}$$

where $B$ and $\lambda$ are the breathing rate ($m^3$/hr) of susceptible students and the viral load or density (copies/$m^3$), expressed as the number of viral copies per unit volume of sputum. $V_B$ was designated as a rectangular box with its long edge oriented in the direction from the nose of the susceptible student toward the ground. The probability of infection was evaluated by tracking droplets in their inhalation zones. The injection droplet volume that enters the inhalation zone $V_B$ was the average inhalation rate of sputum droplets, represented by the initial droplet volume at distance $x$. To analyze the probability of infection in students, $T$ was set as the sum of conversation durations between pairs of students across all breaks in a day.

## Micro-simulation

To analyze virus transmission through contact with virus-carrying students in elementary schools, a micro-simulation was conducted. Micro-simulation is a simulation technique that predicts dynamic changes in societies, economies, cities, and organizations by replicating individual behaviors based on actual behavioral data using probabilistic models. In this study, the contact history recorded on video was replicated, and scenarios in which virus-carrying students were present within this history were simulated. This approach enabled the evaluation of virus transmission in specific situations.

Break time was selected as the contact behavior. Two cases of virus reduction were identified in the contact history of the students: handwashing using water in the handwashing area and disinfection using a disinfectant. In the micro-simulation, it was ensured that the number of virus copies on students' hands would be reduced by $1/10^2$ and $1/10^4$ when washing hands with water and when disinfecting them, respectively [40, 41].

The virus transmission probability between contact materials and viruses and the approximate expression for the decreasing effect of multiple contacts were determined using values reported in previous studies [42, 43]. The items and materials used in this study are listed in S1 Table. The probability of virus transmission was calculated using the mean values from previous studies [42, 43]. These values were obtained through *in vitro* experiments using the influenza virus as a proxy for envelope viruses and a skin model made of protein leather, along with pieces of stainless steel, polypropylene, pottery, wood (veneered board), cardboard, wallpaper, cotton cloth, and model skin. If $n$ denotes the number of contacts for a student, the number of virus copies adhering to the student's hand and the contact item, denoted as $v_n^h$ and $v_n^m$, respectively, vary according to the following equations:

$$v_n^h = (1 - p_{\text{from\_skin}} \times r_d)v^h + v^m \times p_{\text{to\_skin}}, \tag{5}$$

$$v_n^m = (1 - p_{\text{to\_skin}})v^m + v^h \times p_{\text{from\_skin}} \times r_d, \tag{6}$$

where $v^h$ and $v^m$, $p_{\text{from\_skin}}$, and $p_{\text{to\_skin}}$ are the current number of virus copies adhering to the hand and item, the virus transmission probability from the hand to the material, and from the material to the hand, respectively. The obtained values of $v_n^h$ and $v_n^m$ transform into $v^h$ and $v^m$ after the contact has ended. In addition, $r_d$ represents the rate of decrease and is represented by

$$r_d = \begin{cases} 1, & n = 1 \\ 0.88(n-1)^{-1.27}, & n > 1 \end{cases} \tag{7}$$

If the number of viruses adhering to the student's hand and the item was less than one, it was considered as zero copies.

In a single simulation, one virus-carrying student was assigned, assuming that the virus present in the saliva of that student's hand was adhered to by coughing or sneezing immediately after the start of the break. Assuming a COVID-19 patient, it was hypothesized that $10^6$ copies of the virus would adhere to the hand [44].

## Results and discussion

### Communication behavior

**Characteristics of communication between students.** The total number of communications and the communication duration during each break are presented in Table 1. The annotated communication behaviors are listed in S2 Table.

The adjacency matrix of one day was generated using all the communication behaviors described in Table 1 (S1 Fig). In addition, a communication network generated from S1 Fig is presented in S2 Fig. The results were analyzed using data that incorporated the three communication types. The adjacency matrix and network indicate the relationships among the students. These results show that the communication duration and number of communications varied across student pairs. Furthermore, it was demonstrated that there are pairs where the roles of initiator and target do not reverse. To analyze the proportion of students pairs engaged in bidirectional communication where they acted as both initiators and targets, the total degree of each student (which represents the degree of the undirected graph) and the degree of bidirectional communication are presented in Fig 3. The arithmetic average proportion of bidirectional communication among the 30 students was 0.498 with a standard deviation of 0.166. These results indicate that there is heterogeneity in the directionality of relationships based on communication among the students.

The degree distributions were investigated to analyze the characteristics of the communication network. Fig 4 presents the degree distributions of both undirected and directed communication networks and the in-degree and out-degree of directed networks. The arithmetic average degrees of the original degree distribution, indicated by the black bar, were 12.1, 18.1, and 8.83 for the undirected, directed, and both in-degree and out-degree networks, respectively. In addition to the original degree distribution, a modified degree distribution was presented, wherein the degrees were grouped sequentially into sets of three starting from zero. The $P(k)$ of the modified degree was the sum of the $P(k)$ values for the three degrees. The representative value for the modified degree was set as the intermediate value among the three degrees. For example, when modifying the original degrees of 9, 10, and 11, the representative value of the modified degree was set to 10. The modified degree distribution resembled a

**Table 1. Basic data of communication behavior.**

| Break name (Date) | Measurement time | Total number of communication | Total communication duration (second) |
|---|---|---|---|
| Before morning homeroom (21-12-2022) | 8:04:51–8:15:46 | 152 | 2505 |
| 1st break (16-12-2022) | 9:22:05–9:26:25 | 31 | 513 |
| 2nd break (16-12-2022) | 10:09:40–10:34:00 | 119 | 2131 |
| 3rd break (20-12-2022) | 11:20:50–11:25:45 | 117 | 3288 |
| 3rd break (21-12-2022) | 11:22:15–11:25:29 | 27 | 379 |
| Lunch break (21-12-2022) | 12:04:27–12:23:57 | 259 | 4837 |
| Before afternoon homeroom (21-12-2022) | 14:01:11–14:13:04 | 51 | 613 |

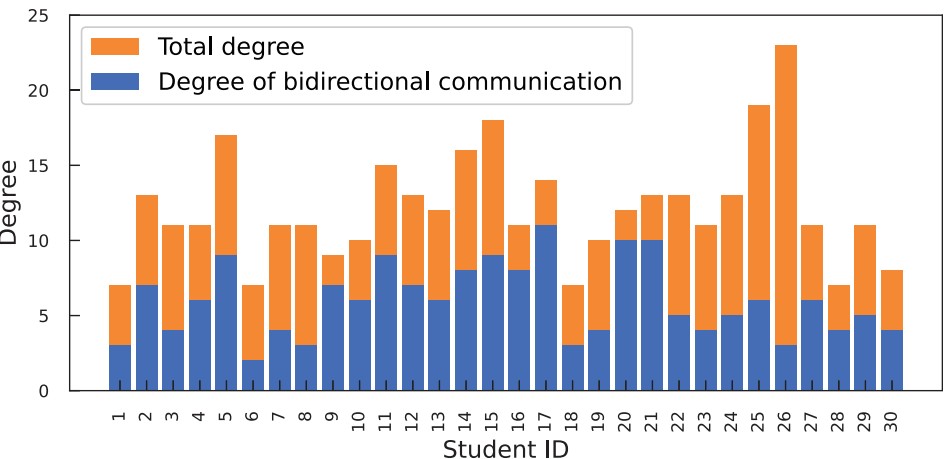

**Fig 3. Proportion of student pairs engaging in bidirectional communication.**

Poisson distribution, where the mode in the modified distribution is considered to be the mean value. These characteristics have also been observed in other elementary school communication network studies as well [22]. However, it was found that the distribution of larger degrees would deviate from the Poisson distribution.

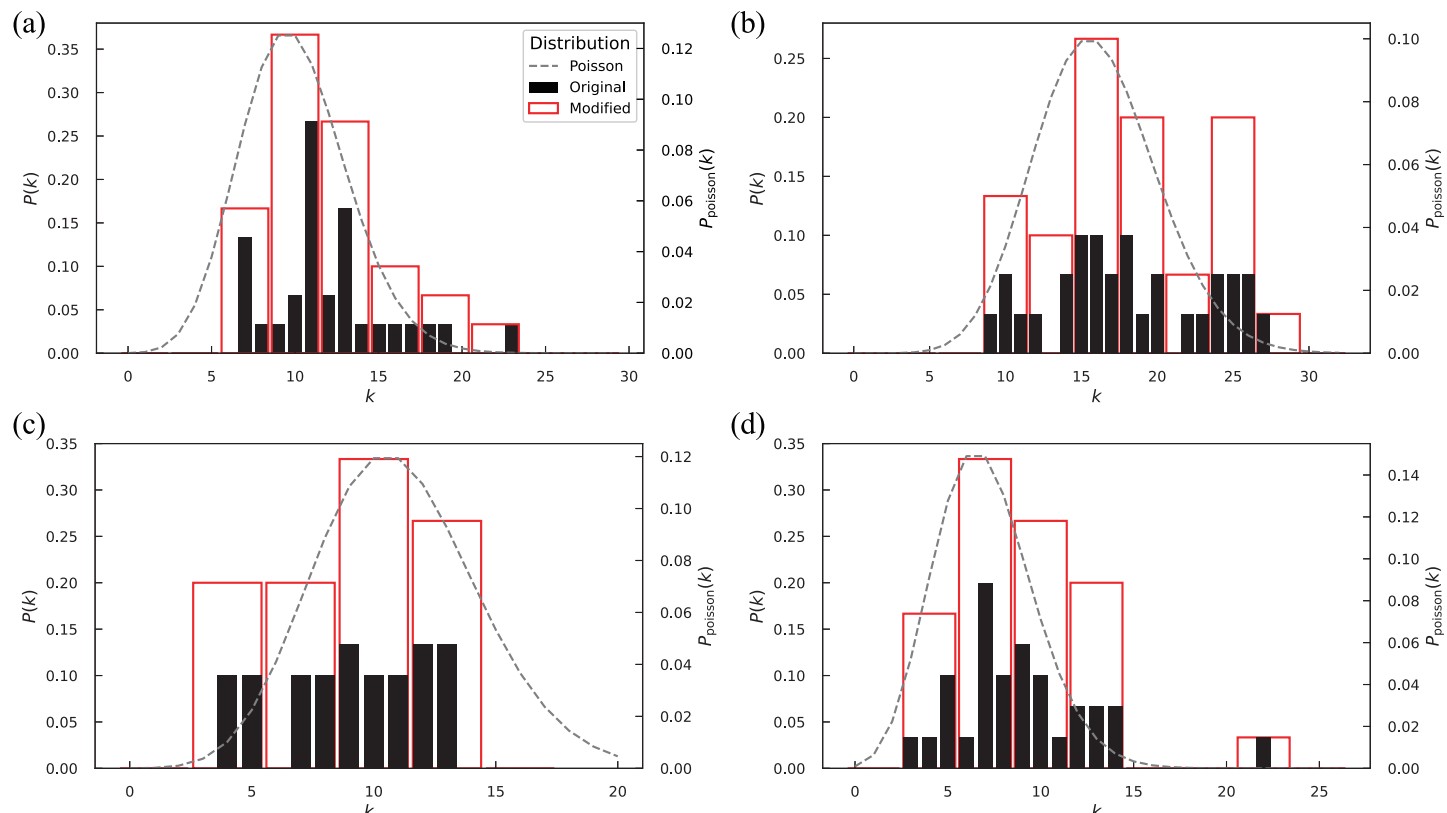

**Fig 4. Degree distribution of communication networks.** (a) Undirected network, (b) Directed network, (c) In-degree distribution of the directed network, (d) Out-degree distribution of the directed network. Black and red bars denote the original degree distribution and the degree distribution modified by grouping degrees into three, respectively. The dashed line denotes the Poisson distribution where the mode in the modified degree distribution is considered as the mean value.

The distribution of communication duration and the number of communications are presented in Fig 5. Communication duration was analyzed based on values aggregated every 10 seconds. Representative values were assigned; for example, 10 seconds for the range 1–10 seconds and 20 seconds for the range 11–20 seconds. To analyze conversation duration, the results of communication duration were used as data, excluding the "contact" communication type. Conversely, the number of communications was analyzed using data that incorporated the three communication types. For Fig 5, the distribution follows a power law. These results indicate that most communications were of short duration and low frequency.

The power law distribution was observed in other elementary school communication network studies as well [16, 20, 23]. In previous studies, communication behavioral data were obtained using contact surveys and proximity sensors in schools [16, 45]. These studies have

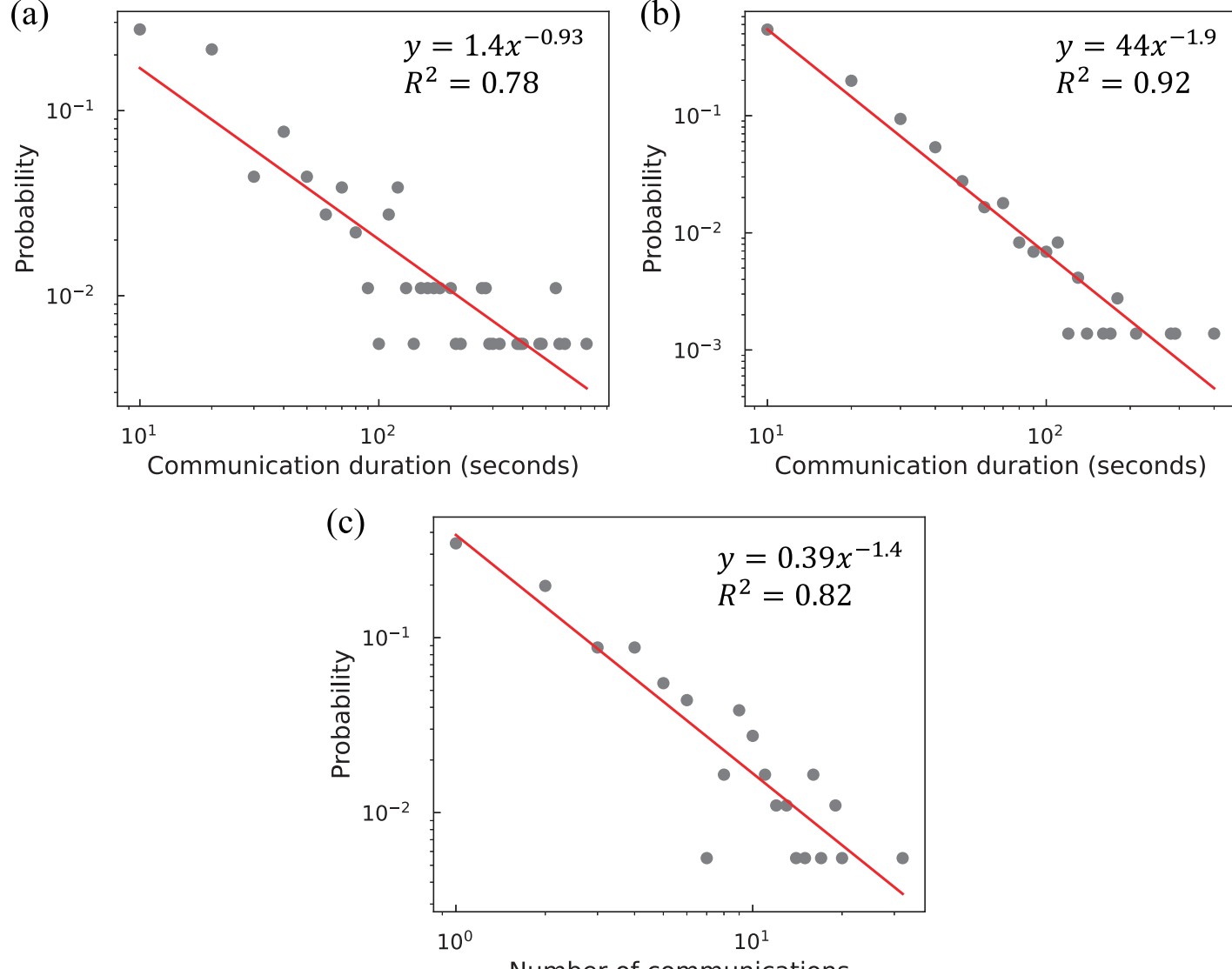

**Fig 5. Distribution of communication duration and the number of communications for the total duration of breaks in a day.** (a) Cumulative communication duration for each student pair, (b) Communication duration of each conversation, (c) Cumulative the number of communications for each student pair. The communication duration was analyzed by aggregating the values every 10 seconds. Red lines denote the approximating curves fitted with a power law.

Table 2. Probability of infection through conversation when $N_0 = 900$. The maximum and minimum probability correspond to $N_0$ of 300 and 2000, respectively.

| Distance ($m$) | $P(\alpha = 1)(\%)$ | $P(\alpha = 5.77)(\%)$ |
|:---:|:---:|:---:|
| 0.5 | 1.29 (0.754–3.10) | 5.05 (2.74–9.62) |
| 1.0 | 0.846 (0.568–1.85) | 3.18 (1.66–6.84) |

shown that while metrics such as the number of communications differ, the structure of communication networks exhibits little variation. Similarly, in this study, which used video-recorded data, the network structure is preserved, consistent with findings from other recording methods.

**Probability of infection through conversation.** The probability of droplet infection through communication was calculated using an equation derived from numerical simulations [34–36]. When a student was a virus carrier, the total communication duration between the virus-carrying student and the susceptible student was used. As there were 30 patterns of infection probabilities from the virus-carrying students to the other 29 susceptible students, a total of 870 infection probabilities were calculated. Data on communication duration excluding the "contact" communication type (conversation duration) were used. The factor that regulated the infectivity caused by variant strains $\alpha$ was set to 1 or 5.77, as described in a previous study [36].

The arithmetic advantages of the probability of infection are listed in Table 2. It was suggested that increasing the distance between virus-carrying and susceptible students during conversations reduced the infection probability. By quantifying the probability of susceptible students becoming infected when one of their classmates was a virus-carrying student, it was possible to assess the likelihood of transmission. Questionnaires and wearable sensor devices could not measure the actual conversation duration of students [11, 16, 20, 23, 24]. By using video-recorded data, it became possible to evaluate the droplet transmission risks.

## Contact behavior

**Characteristics of contact items.** The cumulative number of contacts and the number of contacted items during each break are presented in Table 3. The annotated contact behaviors are depicted in S3 Table. In the "Owner" column of S3 Table, the student ID was annotated when the contacted item belonged to a student. When the contacted item was a shared item, 0 was annotated.

The number of contacts for each item and item ownership in each break is presented in S4 Table. Excluding the "before afternoon homeroom," desks had the highest number of contacts

Table 3. Basic data of contact behavior.

| Break name (Date) | Measurement time | Number of contacts |
|:---:|:---:|:---:|
| Before morning homeroom (16-12-2022) | 8:04:15–8:18:27 | 1412 |
| 1st break (16-12-2022) | 9:20:21–9:25:55 | 567 |
| 2nd break (16-12-2022) | 10:09:36–10:43:23 | 1084 |
| 3rd break (20-12-2022) | 11:20:35–11:25:59 | 1032 |
| 3rd break (21-12-2022) | 11:20:52–11:25:59 | 604 |
| Lunch break (21-12-2022) | 12:07:51–13:19:58 | 4906 |
| Before afternoon homeroom (21-12-2022) | 14:01:40–14:09:48 | 371 |

during all the breaks. In the "before afternoon homeroom," there was a higher frequency of contact with handbags, school bags, and outerwear as students prepared to go home. Similarly, these items were contacted during the "before morning homeroom" because this was the period of arrival at the elementary school. School bags were rarely contacted as they were typically kept in lockers located at the back of the classroom, excluding arrival and departure. During the "lunch break," there was a higher aggregated number of contacts (Table 3 "Lunch break"). This can be attributed to the longer measurement time and the fact that students collectively served lunch, leading to increased interaction among students. In addition, items such as tableware, milk, and lunch mats, were considered lunch-specific. During the "lunch break," as each student needed to wait until all the classmates had finished eating individually, students came into contact with their tablets. Throughout multiple breaks, desks and shirts had a high frequency of contact with personal ownership (self) and with belongings owned by others (others). Regarding the common items, there was a higher frequency of contact with the door, faucet, desk (teacher), and serving table (i.e., serving table cover) located at the front of the classroom. The door was frequently contacted during movement between the classroom and hallway, while the faucet was contacted for handwashing and drinking water in the handwashing area. The desk (teacher) and serving table had a higher frequency of contact because of the accumulation of students around the front of the classroom, where the teacher was present. Since the "3rd break (20-12-2022)" had a higher number of contacts per unit of time, the top 20 contacted items are shown as representatives (Fig 6). These results suggest the potential transmission of viruses through contact with belongings owned by others or shared items. While the possibility of virus transmission via shared items in elementary schools has been investigated in previous studies [25, 26, 28], this study revealed that belongings owned by others have the potential to contribute to virus transmission.

To elucidate the contact with items owned by others, the number of contacts with items and ownership of each item during group and solo activities were analyzed. The data during

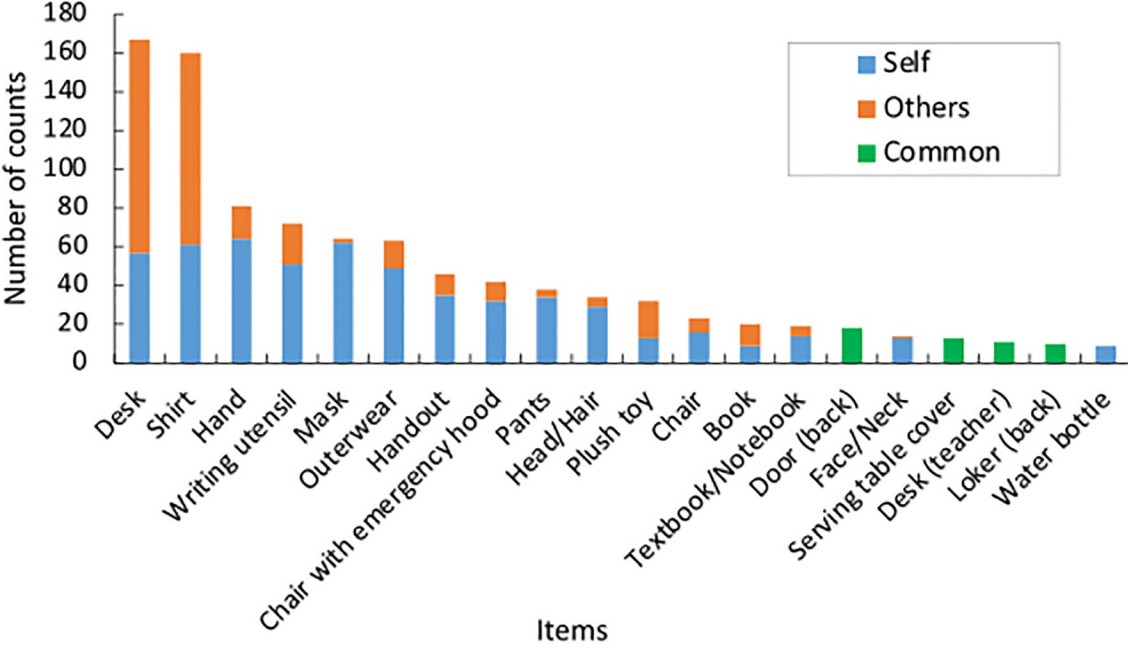

**Fig 6. Top 20 contacted items at the "3rd break (20-12-2022)".**

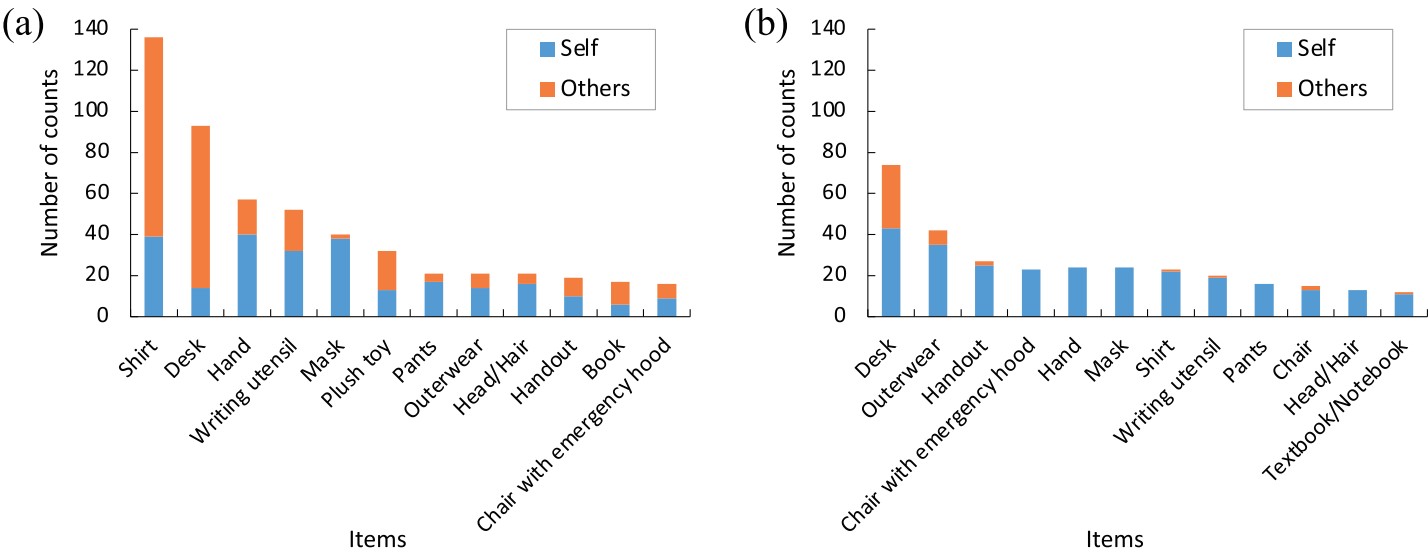

**Fig 7. Contacted items the "3rd break (20-12-2022)".** (a) Group activities, (b) Solo activities. The items with the number of contacts >10 are indicated.

the group and solo activities during each break are presented in S5 and S6 Tables, respectively. As a representative example, the results of the "3rd break (20-12-2022)" are portrayed in Fig 7. Common items were excluded from the analysis. According to these results, students were mostly in contact with belongings owned by others during group activities and with their own items during solo activities. This suggests a potential impact of contact behavior on virus transmission during group activities.

**Network analysis.** To estimate the potential for virus transmission through contact behaviors, contact behaviors were represented using a network. The contact network created from the contact behaviors during the "3rd break (20-12-2022)" is presented in Fig 8.

First, the shortest path between students' hands was investigated. Fig 9 is a red-scale heatmap, which the shortest path between a student ID: $i$ and another student ID: $j$. This heatmap is calculated using the contact network shown in Fig 8. The data represents the number of items required for the virus to transmit from one student's hand to another student's hand. The hand of the starting-point student was not included in the path. Therefore, when the path was 1, it represented the direct connection between the student's hands, and when the path was 2, it indicated the connection through one item. In the contact network, the percentages of paths 1, 2, 3, 4, and 5 were 2.07%, 50.8%, 15.4%, 31.5%, and 0.230%, respectively. Path 2 had the highest occurrence, indicating the potential for virus transmission through a single item in most situations. This suggests that the transmission between student's hands may not require a large number of items.

Next, to predict the candidate items that can be fomites of virus transmission, network metrics such as degree and betweenness centrality were investigated. The top 20 items of the value of degree, excluding students' hands and betweenness centrality, and the degree of only the students' hand in the network are presented in Fig 10. The total values in Fig 10 are presented in S7 Table. As the order of the hands and other items was different, a degree analysis was conducted separately. Incidentally, the degree distribution of the items excluding hands exhibited a power-law-like distribution (S3 Fig). According to Fig 10, common items, such as the serving table cover, desk (teacher), and door (back), had higher degrees and betweenness centrality. Personal belonging items, such as desks, shirts, and hands, exhibited higher degrees and

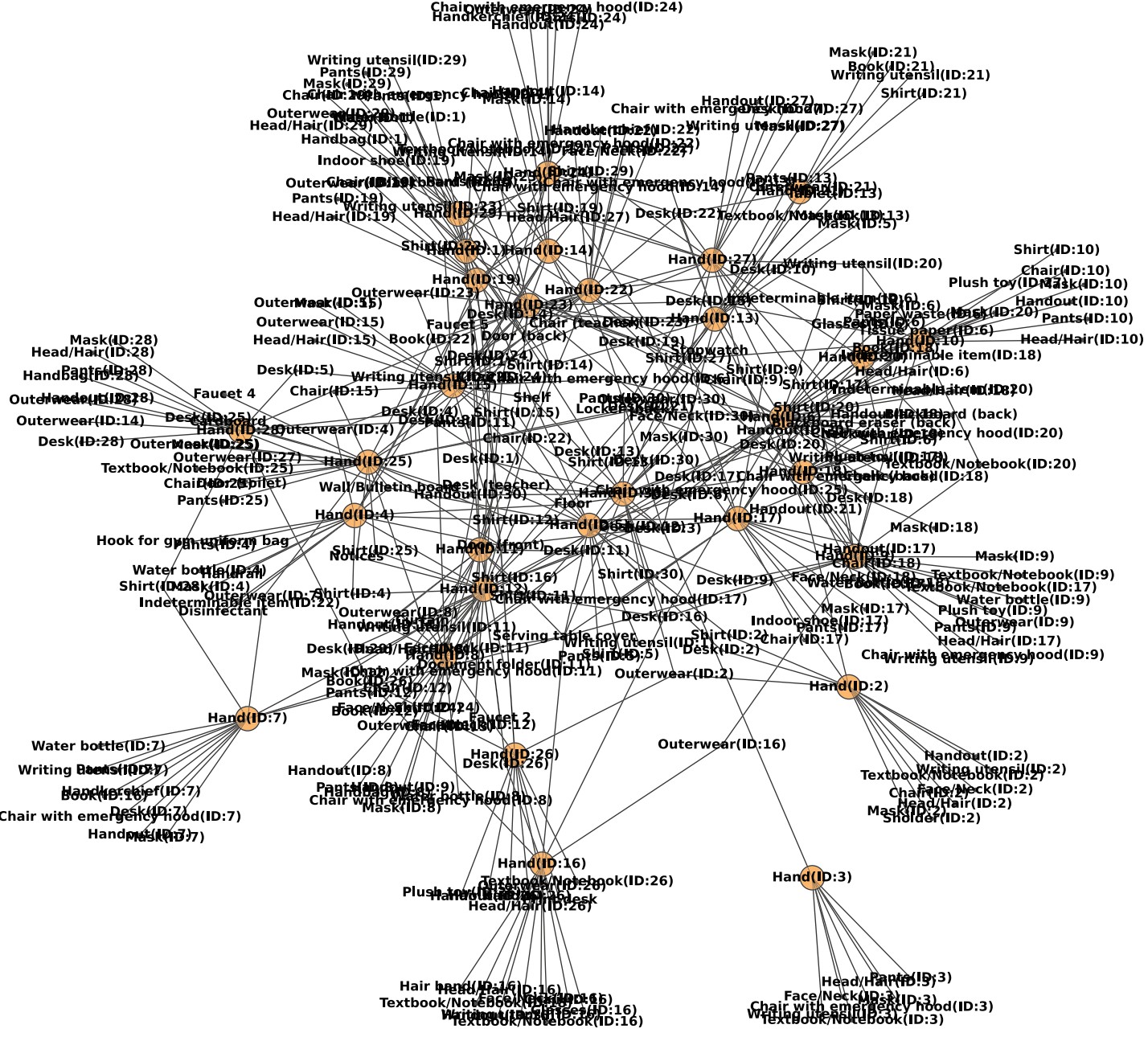

**Fig 8. Network of contact behavior using "3rd break (20-12-2022)".** Orange circles represent students' hands.

betweenness centrality. These items were contacted at a higher frequency, as described in sub-section (Fig 6). These results suggest that these items have the potential to be used as fomites for virus transmission.

**Analysis of contact infection with micro-simulation.** In the previous subsection, potential fomite candidates for virus transmission through contact were predicted based on the contact network. Therefore, in this subsection, we assume the presence of the virus in the actual contact history and analyze the plausibility of the previous subsection using a micro-simulation.

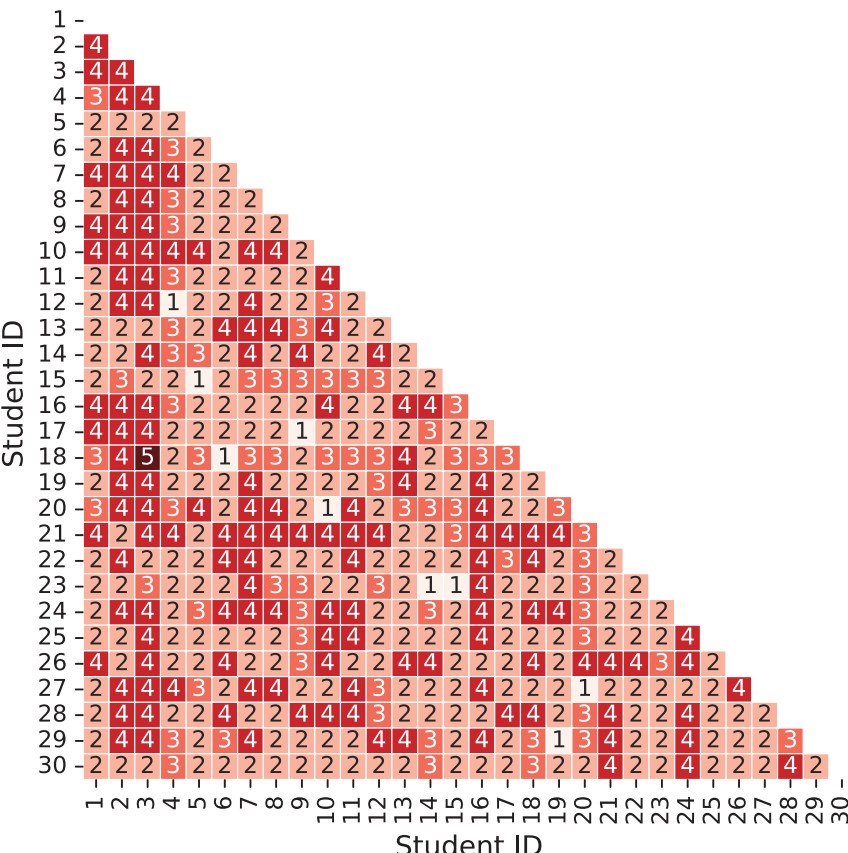

**Fig 9. Heatmap of the shortest path between student's hands.** The numerical values within the heatmap represent the shortest path between students' hands. The hand of the starting-point student is not included in the count.

A micro-simulation was conducted for 30 individuals, assuming one student to be a virus carrier. The contact history of "3rd break (20-12-2022)" was selected as the contact behavior because all students' hands were connected through fomites in this history (Fig 9). Cleaning was not conducted during this period and, therefore, was not taken into consideration. In each simulation, the number of virus copies transmitted to items, other than their own belongings (including hands), was recorded and the results are displayed using a box plot (Fig 11). Only the results where the virus adhered to items with $\geq 1$ copies were included in the figure. The results for the belonging of others and common items are presented separately. The number of samples where $\geq 1$ copies were adhered to each item is presented in S8 Table. Fig 11 displays the items that were sampled more than 50 times for belongings of others and more than five times for common items. In the case of belonging of others, since the maximum number of samples was 870 (30 times 29), the number of samples with virus adherence was higher than that of the common items. From Fig 11(a), when the virus adhered to the belongings of others, the median number of virus copies for each item was approximately $10^1$. For desks, shirts, and hands, which were identified as potential fomites in the previous subsection, the 75th percentile value exceeded $10^2$ copies. This suggests that these components were more susceptible to virus adherence. Furthermore, the maximum number of virus copies adhering to these items was greater than that of other items. Consistent with the results of the micro-simulation, swab sampling surveys conducted in elementary schools have detected viral DNA and RNA on

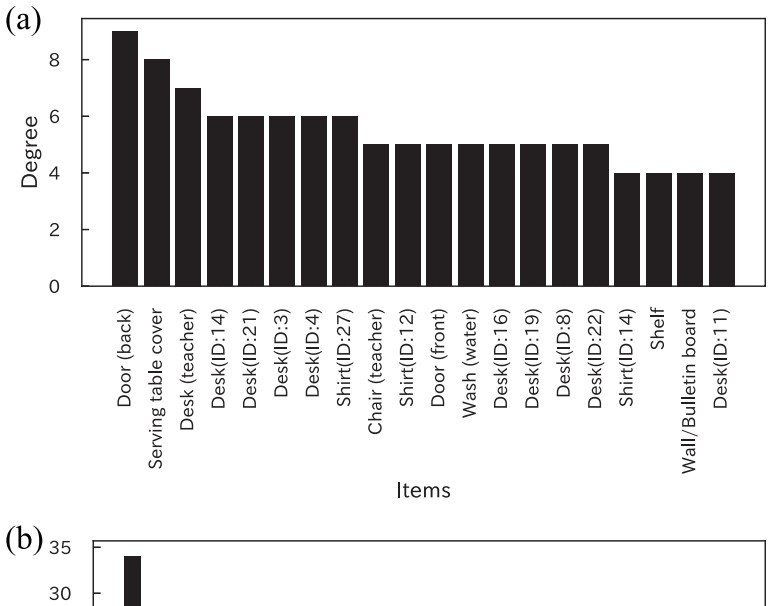

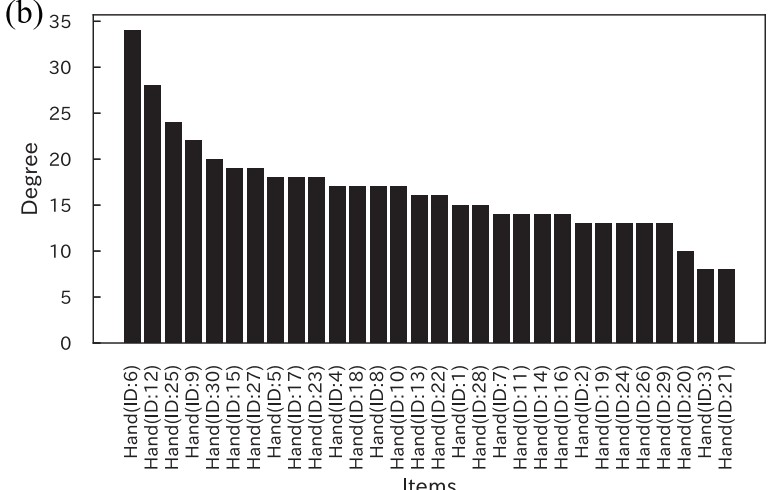

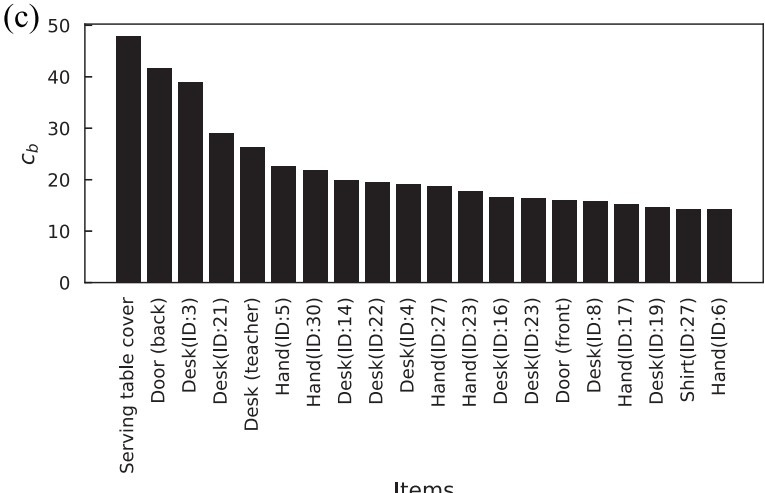

**Fig 10. Network metrics of the network of contact behavior.** (a) Degree of top 20 items without students' hands, (b) Degree of only students' hands, (c) Betweenness centrality of top 20 items.

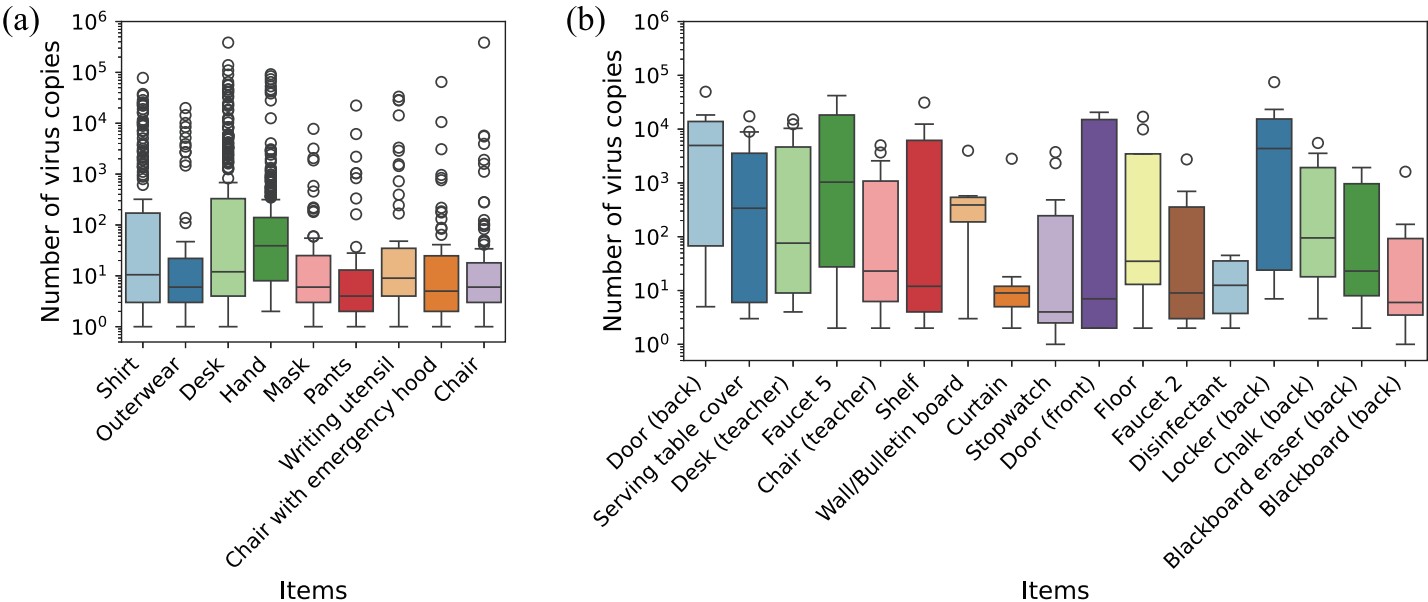

**Fig 11. Relationship between items and adhered virus copies.** (a) Belongings of others, (b) Common items.

desks [25, 29]. In contrast, no previous studies have reported swab sampling of shirts in elementary schools. This is a novel finding regarding items that are likely to have virus adherence. Regarding the common items, although the sample size was smaller, the median number of virus copies surpassed that of the belongings of others. Items, such as doors, serving table covers, and desks (teachers), which were identified as potential fomites, had larger sample sizes and higher 75th percentile (Fig 11(b)). Swab sampling surveys conducted in elementary schools detected viruses and bacteria on doors and faucets [25, 26, 28]. Viruses were detected on doors even when they were often opened [25]. In this contact history, where doors were actively opened and closed, an even higher level of virus adherence may have occurred. Serving table covers and desks (teachers) were predicted as fomites capable of transmitting viruses based on detailed contact behavioral data and the contact networks. These findings suggest that video-recording data collection methods and novel network analysis may provide a evidence-based approach to effective infectious disease countermeasures.

Next, to analyze the relationship between virus transmission and infection, students' hands were examined. Fig 12 illustrates the susceptible students who had the virus adhered to their hands by a virus-carrying student at the end of the simulation. The figure indicates susceptible students with virus adherence of $\geq 1$ copies and $\geq 300$ copies. The value of $\geq 300$ copies represents the minimum average number of virus particles $N_0$. For each virus-carrying student simulation, a contact network was formed based on the items that had virus adherence through contact ("virus transmission contact network" and shown in S4 Fig), and the shortest paths between students' hands were measured. The shortest paths calculated from the virus transmission contact network were presented by color intensity. Path 0 indicates no virus adherence. As the path numbers increase from 1, 2, . . . to 5, it represents that the virus passes through a greater number of items before transmitting from the virus-carrying student to the susceptible student.

The arithmetic average number of susceptible students with virus adherence of $\geq 1$ copies was 19.8 (Fig 12(a)). The maximum was 26 and the minimum was two. Except when ID 5 was

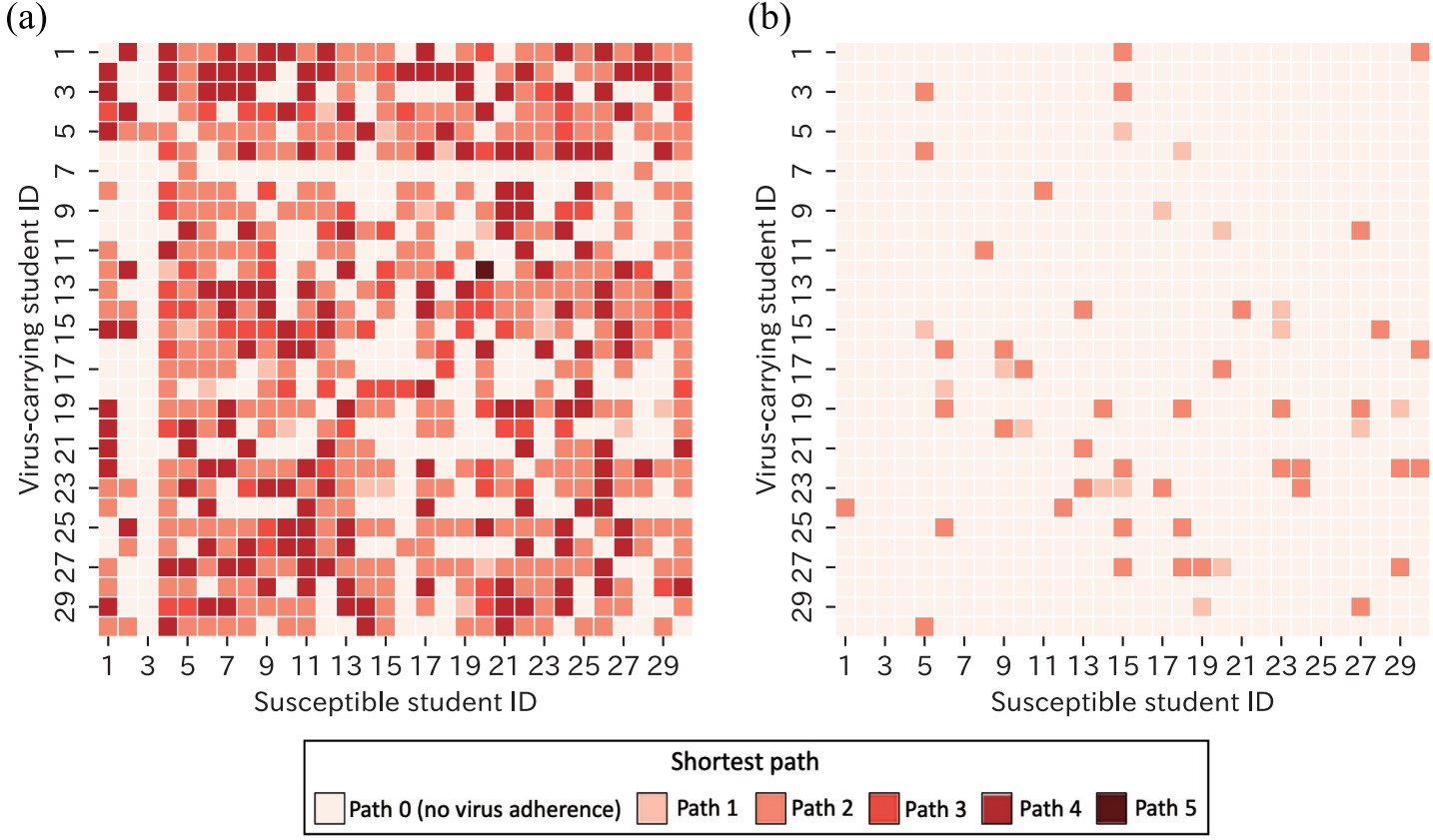

**Fig 12. Heatmap of susceptible students who had the virus adhered to their hands by a virus-carrying student at the final of the simulation.** Susceptible students with virus attachment of (a) $\geq 1$ and (b) $\geq 300$.

a virus carrier, the susceptible hand of ID 3 did not adhere to the virus. This result is consistent with the finding that ID 3 had the highest arithmetic average shortest path for each student (Fig 9). The percentages of the shortest path among all susceptible students with virus adherence were investigated. The percentages of paths 1, 2, 3, 4, and 5 were 3.04%, 54.6%, 12.1%, 30.0%, and 0.169%, respectively (Fig 12(a)). Compared to the distribution of the shortest path in Fig 9, which indicated the potential for virus transmission, the distribution of the virus transmission contact network showed a higher proportion of paths 1 and 2. This suggests that a small number of items is a more efficient transmission pathway. The arithmetic average number of susceptible students with virus adherence of $\geq 300$ copies was 1.93 (Fig 12(b)). The maximum and minimum were five and zero, respectively. In the virus transmission contact network, the shortest path was of two types: path 1 was 27.6% and path 2 was 72.4% (Fig 12 (b)). It was found that the transmission of a high number of virus copies predominantly occurred through a single item. This is attributed to the gradual decay in the number of virus copies being transmitted gradually due to the decreasing rate associated with contact frequencies and low virus transmission probability. It has been demonstrated that in contact-based virus transmission, the presence of a single item in contact with the hands of a virus-carrying student, where a significant amount of the virus has adhered, is crucial.

Finally, the items that transmitted the virus in the micro-simulation were analyzed. The items that transmitted $\geq 1$ or $\geq 100$ copies of the virus to the hands after contact and the number of transmissions by items are presented in Fig 13. The top 20 fomites are indicated in the

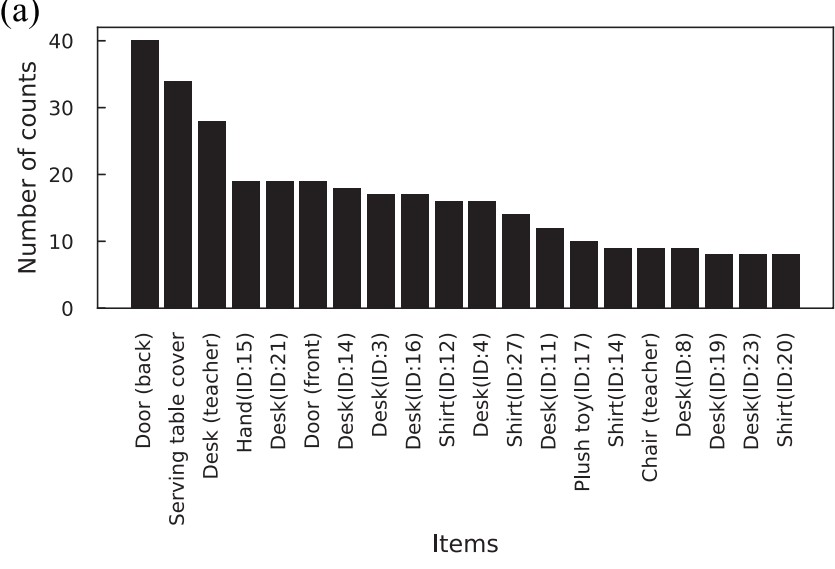

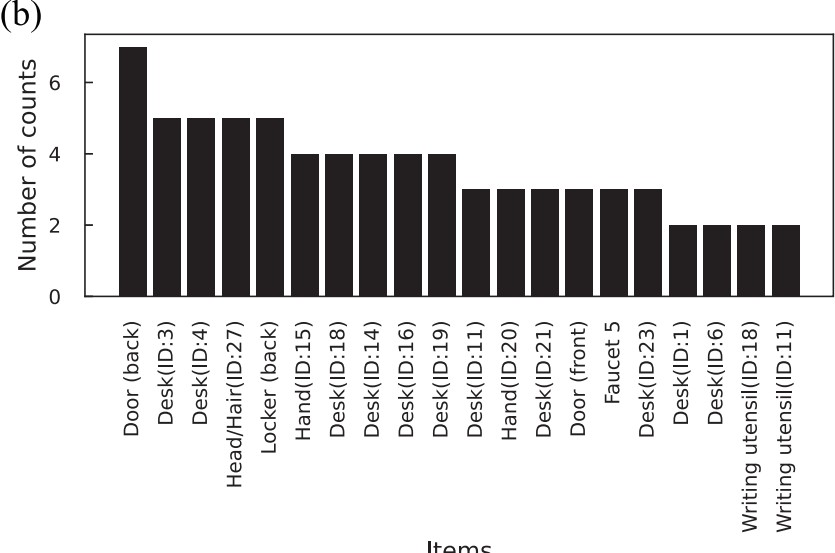

**Fig 13. Fomites and the number of transmission in micro-simulation.** The fomites that transmitted (a) $\geq$ 1 or (b) $\geq$ 100 copies of the virus to the hands after contact. The top 20 fomites are shown.

figure as well. The total relationship between the fomites and the number of transmissions is listed in S9 Table. It should be noted that when virus-carrying student pairs, susceptible students, and items were the same, they were not counted. For instance, if the virus-carrying student ID: $i$ adhered the virus to a door and the susceptible student ID: $j$ contacted the door multiple times, resulting in multiple instances of virus transmission, the number of transmissions by the fomite in that case was considered to be one. In addition, the reason for selecting the threshold of $\geq$ 100 for virus transmission was that it was not possible to explain all the cases where $\geq$ 300 copies of the virus were adhered to the hands of other students using a higher threshold.

In fomites where $\geq$ 1 copies were transmitted, the door, serving table cover, and desk (teacher) had a higher number of counts. This aligns with the results of the analysis of the

contact network's degree and betweenness centrality (Fig 10). Among fomites where $\geq 100$ copies were transmitted, the door had a higher number of counts, while the serving table cover and desk (teacher) did not transmit the virus (S9 Table). This can be attributed to the material of the items. The door, serving table cover, and desk (teacher) were made of stainless steel, cotton cloth, and wood (veneered board), respectively, with average virus transmission probabilities of 0.49%, 0.11%, and 0.20%, respectively. Considering the low virus transmission probabilities of the items other than the doors, it can be concluded that a higher number of virus copies were not transmitted. Furthermore, although the locker (back) had a higher number of counts, it could not necessarily be considered an important item because it was contacted as a single item despite its wide range of contacts. Regarding personal belonging, it appeared that items such as desks, shirts, and hands, which had a high contact frequency and were associated with high network metrics, served as potential fomites.

## Limitations

In this study, communication and contact behaviors were annotated by humans using video recordings. Data accuracy may be reduced in situations where the video quality was poor or where students were clustered closely. Since most students wore masks, the presence or absence of conversations was determined based on the situation; therefore, data reliability was limited.

This study was conducted in a single class in a single elementary school. Thus, the results may be specific to this particular environment. In addition, this study captured behaviors in winter, when the government had issued recommendations regarding COVID-19. Since the COVID-19 pandemic enveloped Japan for approximately three years since its onset in 2020, students had a heightened awareness of infection prevention. Therefore, it might not reflect the natural and naïve behavior of elementary school students. For example, students were consistently wearing masks. There are consistencies with prior studies, however, the results should be treated with caution.

In addition, this study did not include teachers in the analysis. This decision was made because the analysis focused on student relationships who interact as equals, and because teachers were not always present in the areas recorded. However, during the "3rd break (20-12-2022)," multiple students approached the teacher's desk and the serving table at the front of the classroom to meet with the teacher, resulting in a higher contact frequency. Therefore, the teachers may play a role in virus transmission within the classroom.

The contact area was not considered in the micro-simulation. Both virus-carrying and susceptible students contact the same area of each item. However, the contact area would not be limited; therefore, it is important to note that the results of virus transmission might be based on excessive conditions.

## Conclusion

To assess the comprehensive pathogen transmission risk in elementary schools, we employed video recording as a novel approach. This study investigated communication and contact behaviors in an elementary school and analyzed their relevance to droplet and contact transmission. The analysis of communication behaviors revealed the heterogeneous nature of communication among students. The risk of droplet transmission in elementary schools was quantified by calculating droplet transmission probabilities based on conversation duration. The conversation duration, which could not be measured using questionnaires and wearable sensor devices, was obtained through video recording, enabling the evaluation of the probability of infection. In terms of contact behavior, the contact history of elementary school students

was meticulously reconstructed. We established a novel approach to create networks based on this contact history. This enabled the prediction of items with the potential to serve as fomites for viral transmission, as demonstrated by the contact networks. The reliability of these predictions was further supported by the micro-simulations. Items, such as other students' desks, doors, and faucets, which were considered potential fomites, had a significant amount of viral adherence, consistent with swab sampling survey results. In addition, other students' shirts and shared items with high contact frequency and high centrality metrics in the network, which were not evaluated in swab sampling surveys, exhibited substantial viral adherence. These findings suggest that the novel method using detailed contact history and contact networks is a highly effective tool. Furthermore, the results of the micro-simulations indicated that the majority of viral copies were transmitted through single items and did not spread beyond them.

This study examined the behaviors of elementary school students concerning pathogen transmission risk. These insights will contribute to the construction of simulation models for analyzing pathogen transmission risks in elementary schools. We plan to construct models and explore suitable infection control measures in elementary schools.

## Supporting information

**S1 Fig. Heatmaps of adjacency matrix from communication behavior during breaks in a day.** (a) Communication duration, (b) Number of communication between pairs of initiators and targets.
(EPS)

**S2 Fig. Networks from communication behavior of the total duration of breaks in a day.** (a) Communication duration, (b) Number of communication between pairs of initiators and targets. The arrows denote the direction of communication from the initiator to the target. The arrow's thickness in (a) and (b) represents the abundance of communication duration and the number of communications, respectively.
(EPS)

**S3 Fig. Distribution of contacted items excluding hands.** Red and blue lines denote the approximating curves fitted with a power law.
(EPS)

**S4 Fig. Example of virus transmission contact network.** The red and orange circles represent virus-carrying students' hands and the susceptible students' hands with virus adherence of $\geq 1$. The shortest paths between the hands of virus carriers and susceptible students were measured. For instance, the shortest path between ID:28 and ID:7 was path 2.
(EPS)

**S1 Table. Items and materials.**
(XLSX)

**S2 Table. Communication behaviors.**
(XLSX)

**S3 Table. Contact behaviors.**
(XLSX)

**S4 Table. The number of contact for each item and item ownership in each break.**
(XLSX)

**S5 Table. The number of contact during the group activities in each break.**
(XLSX)

**S6 Table. The number of contact during the solo activities in each break.**
(XLSX)

**S7 Table. Network metrics of network of contact behavior.**
(XLSX)

**S8 Table. The number of samples with $\geq 1$ copies.**
(XLSX)

**S9 Table. The relationship between fomites and the number of transmissions.**
(XLSX)

## Acknowledgments

We express our gratitude to the participants, guardians, and teachers who participated in this survey. We would also like to extend our gratitude to Professor Tsubokura and Dr. Rahul from Kobe University and RIKEN for generously providing us with the droplet infection probability function.

## Author Contributions

**Conceptualization:** Setsuya Kurahashi.

**Data curation:** Keisuke Nakajima, Yasuki Kato.

**Formal analysis:** Shuta Kikuchi.

**Funding acquisition:** Setsuya Kurahashi.

**Investigation:** Keisuke Nakajima, Yasuki Kato.

**Project administration:** Takeshi Takizawa, Junichi Sugiyama, Yasushi Kakizawa.

**Supervision:** Takeshi Takizawa, Junichi Sugiyama, Taisei Mukai, Yasushi Kakizawa, Setsuya Kurahashi.

**Visualization:** Shuta Kikuchi.

**Writing – original draft:** Shuta Kikuchi.

**Writing – review & editing:** Shuta Kikuchi, Takeshi Takizawa, Junichi Sugiyama, Taisei Mukai, Yasushi Kakizawa, Setsuya Kurahashi.

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
