## [Decision Letter · Decision Letter 0]

19 Nov 2024

PONE-D-24-31217Quantification of droplet and contact transmission risks among elementary school students based on network analyses using video-recorded dataPLOS ONE

Dear Dr. Kurahashi,

Thank you for submitting your manuscript to PLOS ONE. After careful consideration, we feel that it has merit but does not fully meet PLOS ONE’s publication criteria as it currently stands. Therefore, we invite you to submit a revised version of the manuscript that addresses the points raised during the review process.

We look forward to receiving your revised manuscript.

Kind regards,

Sara Hemati

Academic Editor

PLOS ONE

Journal Requirements:

2. Please expand the acronym “JSPS” (as indicated in your financial disclosure) so that it states the name of your funders in full. 

“This research was supported by JSPS KAKENHI Grant Number 21H01561 and 23H00503A. 

https://kaken.nii.ac.jp/ja/grant/KAKENHI-PROJECT-21H01561/

https://kaken.nii.ac.jp/ja/grant/KAKENHI-PROJECT-23H00503/

This research was supported by grants as joint research from LION Corporation.”

**Additional Editor Comments:**

Reviewer 1

The article is good and will be of benefit to teaming researchers. Unfortunately, the language of the article is not perfect, ‎containing ‎multiple grammar and verbal errors throughout the text, which hampers ‎perception of the ‎work. I recommend that the authors should carefully revise the text for ‎the language style ‎and errors, for which purpose they are advised to consult those who ‎have expertise in ‎writing scientific articles in English.‎

‎The introduction of the manuscript should be analyzed the critical gap in the ‎literature ‎and how the present study mitigates the gap. Also, the objectives are not clearly ‎written ‎and mentioned.‎

‎

‎The abstract and conclusion sections should be improved to show the main ‎research ‎findings.‎

‎‎The authors should put more efforts on the discussion with the help of latest ‎and ‎pertinent references.‎

Reviewer 1

Dear Authors,

First of all, I would like to state that this publication is a very scientific, technical and interesting research.

The text of the article has been uploaded to the system twice. There are referencing deficiencies, typographical errors. The latest version needs to be uploaded.

Line 175-176 also has two communication behavior headings.

It will be of interest to the reader to give brief information in the method section about what kind of research and analysis is the micro-simulation.

It should be noted that the situation of the teacher factor in the classroom was not analyzed in the study, and there was no cleaning in the classroom.

From the contacted materials, it should be indicated whether the backpack is kept in cabinets.

If there is information about how long the cases wore a mask, it should be specified.

It should be noted that the image is not taken in hand washing, toilets and faucets.

In Figure 6, the word numver should be corrected as number.

What is Heatmap? How is it interpreted and what does it mean? A brief description of this topic informs the reader and attracts attention.

Reviewers' comments:

Reviewer's Responses to Questions

**Comments to the Author**

1. Is the manuscript technically sound, and do the data support the conclusions?

Reviewer #1: Yes

Reviewer #2: Yes

2. Has the statistical analysis been performed appropriately and rigorously? 

Reviewer #1: Yes

Reviewer #2: N/A

3. Have the authors made all data underlying the findings in their manuscript fully available?

Reviewer #1: Yes

Reviewer #2: Yes

4. Is the manuscript presented in an intelligible fashion and written in standard English?

Reviewer #1: Yes

Reviewer #2: Yes

5. Review Comments to the Author

Reviewer #1: The article is good and will be of benefit to teaming researchers. Unfortunately, the language of the article is not perfect, ‎containing ‎multiple grammar and verbal errors throughout the text, which hampers ‎perception of the ‎work. I recommend that the authors should carefully revise the text for ‎the language style ‎and errors, for which purpose they are advised to consult those who ‎have expertise in ‎writing scientific articles in English.‎

‎The introduction of the manuscript should be analyzed the critical gap in the ‎literature ‎and how the present study mitigates the gap. Also, the objectives are not clearly ‎written ‎and mentioned.‎

‎

‎The abstract and conclusion sections should be improved to show the main ‎research ‎findings.‎

‎‎The authors should put more efforts on the discussion with the help of latest ‎and ‎pertinent references.‎

‎

Reviewer #2: Dear Authors,

First of all, I would like to state that this publication is a very scientific, technical and interesting research.

The text of the article has been uploaded to the system twice. There are referencing deficiencies, typographical errors. The latest version needs to be uploaded.

Line 175-176 also has two communication behavior headings.

It will be of interest to the reader to give brief information in the method section about what kind of research and analysis is the micro-simulation.

It should be noted that the situation of the teacher factor in the classroom was not analyzed in the study, and there was no cleaning in the classroom.

From the contacted materials, it should be indicated whether the backpack is kept in cabinets.

If there is information about how long the cases wore a mask, it should be specified.

It should be noted that the image is not taken in hand washing, toilets and faucets.

In Figure 6, the word numver should be corrected as number.

What is Heatmap? How is it interpreted and what does it mean? A brief description of this topic informs the reader and attracts attention.

Thank you. Congratulations.

6. PLOS authors have the option to publish the peer review history of their article (what does this mean?). If published, this will include your full peer review and any attached files.

Reviewer #1: No

Reviewer #2: **Yes: **ERSIN DEMIRER

---

## [Author Response · Author response to Decision Letter 0]

18 Dec 2024

Dear Reviewers:

We sincerely thank you for providing thoughtful and detailed comments on our manuscript. We are deeply grateful for the constructive suggestions, which have greatly contributed to improving the quality of our work.

Our responses to the Reviewers' Comments have been uploaded to the "Attach Files" section of the Editorial Manager system. Additionally, Reviewer 1 provided comments regarding grammar and verbal errors. To address these concerns, we have revised the manuscript using a professional English proofreading service. The certificate of editing has been uploaded as "Certificate_of_editing" in the "Attach Files" section.

We would greatly appreciate it if you could kindly review the revised manuscript at your convenience. Thank you for your time and effort.

Best regards,

Setsuya Kurahashi

---

## [Editor Report · Decision Letter 1]

26 Dec 2024

Quantification of droplet and contact transmission risks among elementary school students based on network analyses using video-recorded data

PONE-D-24-31217R1

Dear Dr. Kurahashi,

We’re pleased to inform you that your manuscript has been judged scientifically suitable for publication and will be formally accepted for publication once it meets all outstanding technical requirements.

Kind regards,

Sara Hemati

Academic Editor

PLOS ONE
---

## [Editor Report · Acceptance letter]

20 Jan 2025

PONE-D-24-31217R1 

PLOS ONE

Dear Dr. Kurahashi, 

I'm pleased to inform you that your manuscript has been deemed suitable for publication in PLOS ONE. Congratulations! Your manuscript is now being handed over to our production team.

Kind regards, 

on behalf of

Dr. Sara Hemati 

Academic Editor

PLOS ONE